# Fast and Sustained Axonal Growth by BDNF Released from Chitosan Microspheres

**DOI:** 10.3390/md21020091

**Published:** 2023-01-27

**Authors:** Inmaculada Aranaz, Niuris Acosta, Julia Revuelta, Agatha Bastida, Víctor Gómez-Casado, Concepción Civera, Leoncio Garrido, Eduardo García-Junceda, Ángeles Heras, Andrés R. Alcántara, Alfonso Fernández-Mayoralas, Ernesto Doncel-Pérez

**Affiliations:** 1Departamento de Química en Ciencias Farmacéuticas, Facultad de Farmacia, Universidad Complutense de Madrid, Plaza de Ramón y Cajal s/n, 28040 Madrid, Spain; 2Departamento de Química Bio-Orgánica, Instituto de Química Orgánica General (IQOG-CSIC), CSIC, Juan de la Cierva 3, 28006 Madrid, Spain; 3Laboratorio de Química Neuro-Regenerativa, Hospital Nacional de Parapléjicos, SESCAM, Finca la Peraleda s/n, 45071 Toledo, Spain; 4Departamento de Química-Física, Instituto de Ciencia y Tecnología de Polímeros (ICTP-CSIC), CSIC, Juan de la Cierva 3, 28006 Madrid, Spain

**Keywords:** brain-derived neurotrophic factor, chitosan, microspheres, dynamic light scattering, PC12 cells, zeta potential

## Abstract

Brain-derived neurotrophic factor (BDNF) regulates dendritic branching and dendritic spine morphology, as well as synaptic plasticity and long-term potentiation. Consequently, BDNF deficiency has been associated with some neurological disorders such as Alzheimer’s, Parkinson’s or Huntington’s diseases. In contrast, elevated BDNF levels correlate with recovery after traumatic central nervous system (CNS) injuries. The utility of BDNF as a therapeutic agent is limited by its short half-life in a pathological microenvironment and its low efficacy caused by unwanted consumption of non-neuronal cells or inappropriate dosing. Here, we tested the activity of chitosan microsphere-encapsulated BDNF to prevent clearance and prolong the efficacy of this neurotrophin. Neuritic growth activity of BDNF release from chitosan microspheres was observed in the PC12 rat pheochromocytoma cell line, which is dependent on neurotrophins to differentiate via the neurotrophin receptor (NTR). We obtained a rapid and sustained increase in neuritic out-growth of cells treated with BDNF-loaded chitosan microspheres over control cells (*p* < 0.001). The average of neuritic out-growth velocity was three times higher in the BDNF-loaded chitosan microspheres than in the free BDNF. We conclude that the slow release of BDNF from chitosan microspheres enhances signaling through NTR and promotes axonal growth in neurons, which could constitute an important therapeutic agent in neurodegenerative diseases and CNS lesions.

## 1. Introduction

Neurodegenerative diseases are common causes of morbidity and cognitive decline in the elderly, and they include Alzheimer’s disease, Parkinson’s disease, amyotrophic lateral sclerosis, Huntington’s disease, etc. This diverse set of diseases is characterized by the slow loss of anatomically or physiologically relevant neural systems [1]. 

Brain-derived neurotrophic factor (BDNF) is a member of neurotrophic factors family, which includes nerve growth factor (NGF), neurotrophin-3 (NT-3) and neurotrophin-4 (NT-4). BDNF is the most abundant neurotrophic factor in the brain, expressed in several brain regions including the cortex and hippocampus, and it exerts beneficial effects in the development, survival and maintenance of neurons in the central nervous system (CNS) [2]. 

BDNF, like other neurotrophic factors, has high clearance and a very short plasma half-life, due to the rapid degradation by extracellular peptidases or significant receptor-mediated clearance [3,4]. In general, target tissue concentrations of these injected proteins in the systemic circulation are lower than in plasma, because the rate of entry into the tissue interstitial space will be less than the rate of exit through tissue lymphatic drainage [3,4].

A low level of BDNF in peripheral blood or CNS and insufficient transformation of pro-BDNF into mature BDNF (mBDNF) have been found to be involved in the pathogenesis of neurodegenerative diseases [5,6]. In these diseases, failure to switch from pro-BDNF to mBDNF is due to abnormal proteolytic cleavage. For example, altered expression and/or activities in components of the tPA/plasmin system (extracellular cleavage proteases) have been implicated in pathological processes related to depression and anxiety [6]. Normal circulating mBDNF levels are critical not only in neurodegenerative conditions in the elderly, but also during the initial stages of human life. A decreased half-life of circulating mBDNF has been correlated with a poor recovery from some stressful psychological events during childhood or adolescence [7].

Chitosan is a copolymer of N-acetylglucosamine and glucosamine obtained from the deacetylation of chitin, a structural component of arthropod exoskeletons, fungal cell walls or fish scales, and it has several applications in biomedicine [8]. Some chitosan formulations have been used to encapsulate drugs and evade phagocytic uptake by reducing opsonization by blood proteins, thereby increasing drug bioavailability [9]. In addition, chitosan-containing composite complexes enhanced drug release, antimicrobial activity and cell viability [10,11]. 

The aim of this study was to encapsulate mature BDNF in chitosan microspheres to protect it from possible hydrolysis or degradation, without affecting neurotrophin biological activity. Neuritic growth in PC12 cells was selected as a practical and specific approach for the detection of neural differentiation and axonal promoting activity after BDNF stimulation [12]. This work tests the effective release of neurotrophin in PC12 cells after encapsulation in chitosan microspheres, prior to animal experimentation in preclinical studies for neurological diseases.

## 2. Results

### 2.1. Chitosan Microspheres-BDNF versus Chitosan Hydrochloride Microspheres-BDNF 

Unmodified chitosan and chitosan hydrochloride were used as raw materials for BDNF encapsulation in microspheres. The use of unmodified chitosan yielded some aggregation of microspheres after the encapsulation process (Figure 1a,b). In contrast, the use of hydrochloride chitosan at the beginning of encapsulation produced individualized microspheres (Figure 1c,d).

### 2.2. Zeta Potential and Dynamic Light Scattering of Chitosan Microspheres

The zeta potential (ζ) is an indicator of the stability of colloidal dispersions [13]. The magnitude of ζ indicates the degree of electrostatic repulsion between adjacent, similarly charged particles in a dispersion. The dispersion of chitosan microspheres-BDNF shows a lower zeta potential (ζ = −9,7 ± 10 mV) than chitosan hydrochloride microspheres-BDNF (ζ = −70.7 ± 8.7 mV), and the size of particles measured by dynamic light scattering (DLS) was 10 times higher in chitosan microspheres-BDNF (6091 nm) than in chitosan hydrochloride microspheres-BDNF (508 nm). Salt powder was used as a control for comparison with chitosan particles (Table 1).

The low size and high negative superficial charge in the chitosan microspheres were consistent with individual microspheres captured in SEM images (Figure 1d). Some flocculation was observed in chitosan microspheres-BDNF (Figure 1a,b), which could be derived from the glue properties described for chitosan biomaterial [14,15]. This event did not affect the activity of neurotrophin in chitosan microspheres-BDNF, as seen below. 

### 2.3. Effect of Chitosan Microspheres-BDNF on Neural Cell Differentiation and Neuritic Growth in PC12 Cells

The activity of encapsulated BDNF was observed in rat pheochromocytoma cell line PC12, which is dependent of neurotrophins for differentiation via the neurotrophin receptor (NTR); it was detected by a neuronal specific marker, neurofilament. Free BDNF or BDNF encapsulated in chitosan microspheres were assayed at 72 h and 120 h, whereby empty chitosan capsules were used as a control (Figure 2), without affecting the viability of cells (see Appendix A). The majority of PC12 cells in the absence of neurotrophin remain in the rounded form (Figure 2a,d); cell differentiation was observed in the presence of BDNF (Figure 2b,c,e,f). We obtained a very significant increase (*p* < 0.001) in neuritic out-growth for encapsulated BDNF-treated cells over control cells (Figure 2g). The average of distance was used to calculate neuritic growth velocity, which was three times higher in encapsulated BDNF than in free BDNF at 72 h or 120 h over control cells (Figure 2h).

## 3. Discussion

The brain-derived neurotrophic factor (BDNF) is related to recovery after a disease or injury in the central nervous system (CNS) [16,17,18]. The axonal growth is diminished after a deleterious process in CNS [19], and exogenous administration of BDNF could promote axonal elongation [20]. We selected chitosan microspheres as the carrier of this neurotrophin because of their safety in CNS [12] and capability in preserving the BDNF protein from an adverse environment in the neuroinflammation process after CNS injury [19], and to guarantee a local release of the molecule.

BDNF binds to tropomyosin-related kinase receptor type B (TrkB) and p75 neurotrophin receptor (NTR); both receptors can be found in diverse types of cells and BDNF signal transduction takes place with diverse effects [21,22,23]. The PC12 cells bear NTR receptor in the cell surface and these have been used as a model for BDNF signaling, cell differentiation and neuritic out-growth [24,25,26]. We observed that BDNF released from chitosan microspheres produced a higher neural cell differentiation and neuritic out-growth than free BDNF in PC12 cells, in a significant manner. We did not obtain this effect in BNDF-chitosan hydrochloride microspheres (data not shown), even though these were obtained with a better dispersion in solution. This high neural cell differentiation and neuritic out-growth could be attributed to the glue properties of chitosan that group chitosan microspheres-BDNF [14,15]; consequently, the BDNF molecules became more abundant than free BDNF and produced a rapid and sustained axonal growth. 

There was a reduction in neuritic growth rate in the case of chitosan-BDNF microspheres, but not for the BDNF-free treatment (Figure 2h). The kinetic release of proteins from the chitosan particles reaches a maximum around the third day and is maintained on successive days [27,28]. Accordingly, a high concentration of local BDNF released from chitosan particles promotes the rate of neuritic growth in PC12 cells; however, these cells are close to the growth limit, and, thus, the growth rate decreases. In the case of free BDNF, the rate of neuritic growth is almost the same because the BDNF concentration did not change in solution during the assay.

It has been reported that the use of chitosan microspheres containing BDNF supports the survival of neurons and the out-growth of neurites in a similar way to free BDNF [29]. We speculate that the axonal-promoting effect of free BDNF in nerve tissues could be halted by a clearance of neurotrophins and metabolization by phagocytic or glial cells, avoiding reaching neurons. In contrast, here we have shown for the first time that BDNF release from chitosan microspheres is concentrated near target cells, improving the event of BDNF/NTR interaction and increasing the velocity and distance of axonal growth. 

## 4. Materials and Methods 

### 4.1. Cell Line

Rat pheochromocytoma, PC12 cells [PC-12 (ATCC^®^ CRL-1721™)], were used as the cell model for neural cell differentiation and neuritic out-growth. The PC12 cells were cultured in Dulbecco’s Modified Eagle Medium, DMEM (Lonza, Cultek, Madrid, Spain) plus 7.5% horse serum and 7.5% bovine serum until treatment with encapsulated BDNF. 

### 4.2. Encapsulation of BDNF in Chitosan Microspheres

About 50 mg of chitosan (Chitofarm S, Chitinor, Norway) with a degree of deacetylation of 84% and a molecular weight (Mw) of 84.5 kDa or chitosan hydrochloride (Protasan UP CL 113, Novamatrix, Norway) with a degree of deacetylation of 90% and a molecular weight (Mw) of 200 kDa was dissolved in 50 mL of 0.3 M acetic acid (Panreac)/0.2 M sodium acetate (Panreac) buffer overnight at RT. The polymer solution was filtered through a 5 μm pore filter and 1000 ng of brain derived neurotrophic factor, BDNF (Peprotech, Cranbury, NJ, USA), at 50 ng/µL in DMEM (Lonza, Cultek, Madrid, Spain) was added. This mixture was stirred for 10 min until homogenization. After homogenization, solutions were spray-dried using a mini spray dryer B-290 (Büchi Labortechnik AG, Flawil, Switzerland). The conditions selected for the process were an aspirator efficiency of 80%, a pump power of 20%, and an inlet air temperature of 140 °C. The atomized product was a white powder that was stored in a desiccator at room temperature and analyzed by scanning electron microscopy (SEM), zeta potential (ζ) and dynamic light scattering (DLS).

### 4.3. Analysis by Zeta Potential (ζ) 

Zeta potential analyses were performed using a Zetasizer Nano ZS (Malvern Instruments, Worcestershire, UK). The zeta potential measurements were performed at 25 °C, the equilibration time was fixed at 120 s and the measurements were carried out in automatic mode (a minimum of 10 runs and a maximum of 100 runs). Calculations were carried out using the Smoluchowski approximation. Prior to the measurement, the samples were re-suspended in 1 mM sodium chloride, NaCl (Sigma-Aldrich Chemie, Steinheim, Germany), and loaded into a folded capillary cell (DTS 1070). 

### 4.4. Analysis by Dynamic Light Scattering (DLS)

Particle size distribution of the nanoparticles was analyzed by DLS with Zetasizer Nano ZS (Malvern Instruments, Worcestershire, UK). A measurement angle of 173° (Backscatter) and automatic measurement duration was selected. Samples were re-suspended in PBS (pH 7.4) and loaded into a disposable cuvette.

### 4.5. Chitosan Microspheres-BDNF Treatment 

Chitosan microsphere treatment of PC12 cells followed a previous report by our group [30], with minor variations. In brief, PC12 cells were seeded at 50,000 cells per well and incubated for 24 h at 37 °C/5% CO_2_ to facilitate cell adherence. Then, the medium was removed and chitosan-loaded BDNF microspheres (1 mg/mL) dissolved in DMEM at 2% bovine serum were added. Free BDNF (100 ng/mL) and uncharged chitosan microspheres (1 mg/mL) were used as controls. Cells were fixed at the 3rd and 5th days of cell culturing.

### 4.6. Immunocytochemistry 

The fixed cells were immunostained with primary mouse monoclonal antibody for neurofilament (Enzo Life Sciences, New York, NY, USA), diluted 1/500. After repeated washing with phosphate-buffered saline (PBS), the cells were treated with a secondary antibody, Alexa 594 conjugated goat antimouse (Molecular Probes, Life Technologies, USA) at 1/1000, and Hoechst agent (Sigma-Aldrich Chemie, Steinheim, Germany) at 10 μg/mL was added for cell nuclei labeling. The assays were made three times, and in each replicate, 40 fields were selected and analyzed by IN Cell Analyzer 1000 Cellular Imaging (General Electric).

### 4.7. Statistical Analysis

Data from 3 groups were compared using 2-way ANOVA followed by Bonferroni’s posttests. The continuous variables were expressed as mean ± standard deviation. A *p*-value of <0.05 was considered significant. All statistical analyses were carried out using GraphPad Prism, version 5.

## 5. Conclusions

The promotion of axonal growth by BDNF released from chitosan microspheres was revealed. BDNF released from the chitosan microspheres produced greater neural cell differentiation and neuritic growth than free BDNF. The intrinsic adhesive properties of the chitosan microspheres and the slow local release of neurotrophin supported the enhancement of the neuritic growth process in PC12 cells. This could open new therapeutic avenues for the use of BDNF-loaded chitosan microspheres in models of CNS pathology, such as Parkinson’s, Alzheimer’s disease or spinal cord injury therapies.

## Figures and Tables

**Figure 1 marinedrugs-21-00091-f001:**
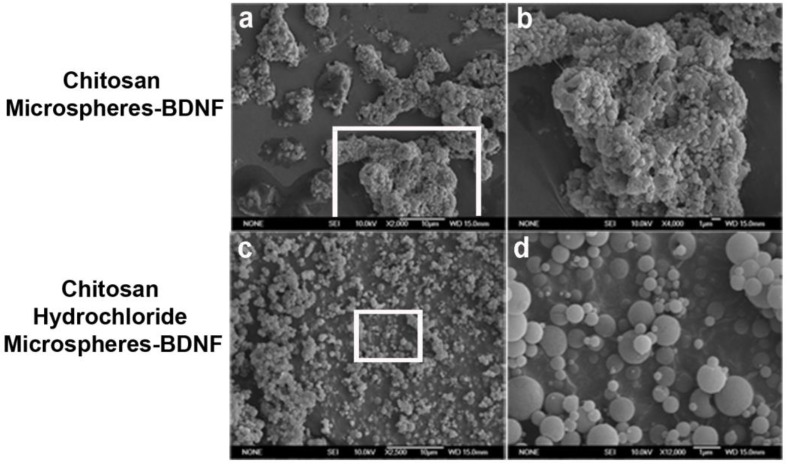
Scanning electron microscopy analysis of chitosan microspheres containing BDNF. After the BDNF encapsulation process, unmodified chitosan microspheres showed some aggregation (**a**,**b**). In contrast, individual BDNF-chitosan hydrochloride microspheres were obtained under the same conditions (**c**,**d**). Magnification bars at 10 µm (**a**,**c**) and 1 µm (**b**,**d**).

**Figure 2 marinedrugs-21-00091-f002:**
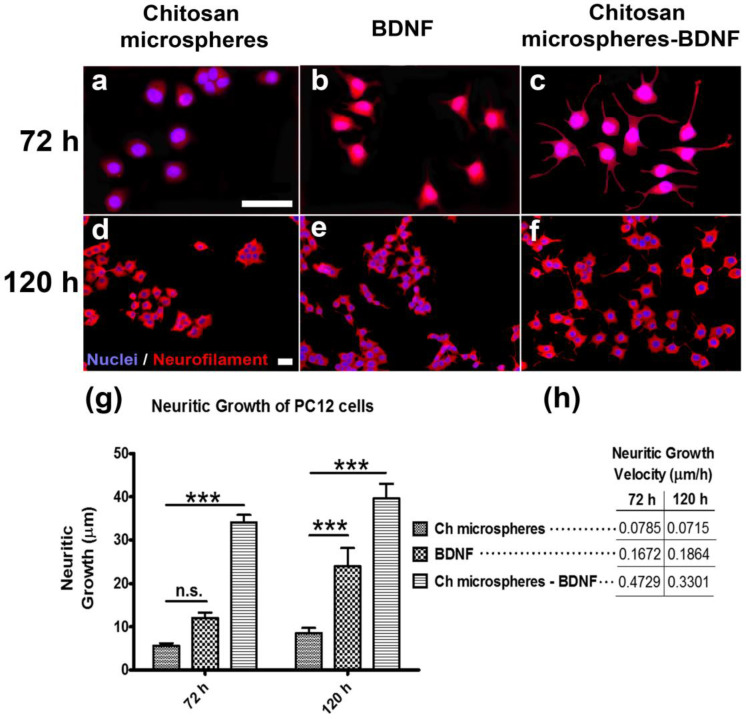
Axonal growth promoting BDNF release from chitosan microspheres. The PC12 cells were cultured at 72 h and 120 h in the presence of chitosan microspheres (**a**,**d**), free BDNF (**b**,**e**) or chitosan microspheres containing BDNF (**c**,**f**). The neuritic out-growth in PC12 cells was revealed by anti-neurofilament in red, and nuclei by Hoechst in blue (see (**d**)). The distance (**g**) and velocity of axonal growth (**h**) were significantly higher in chitosan microspheres-BDNF than other experimental variants at 72 h and 120 h. Ch, chitosan; scale bars correspond to 50 µm and 30 µm for (**a**–**c**) and (**d**–**f**), respectively; *** *p* < 0.001.

**Table 1 marinedrugs-21-00091-t001:** Zeta potential and dynamic light scattering of chitosan microspheres containing BDNF.

Sample	ζ ^1^ (mV)	Size by DLS ^2^ (nm)
Chitosan Microspheres-BDNF	−9.7 ± 10	6091
Chitosan Hydrochloride Microspheres-BDNF	−70.7 ± 8.7	508
Salt Powder (Control)	17.5 ± 3.2	5800

^1^ Zeta potential; ^2^ dynamic light scattering.

## Data Availability

The original contributions presented in the study are included in the article/Appendix A; further inquiries can be directed to the corresponding author.

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
