# Peer review of "Fast and Sustained Axonal Growth by BDNF Released from Chitosan Microspheres"

_marinedrugs, 2023, doi:10.3390/md21020091_

Round 1

Reviewer 1 Report

In this research article, the authors presented “Fast and Sustained Axonal Growth by BDNF Released from Chitosan Microspheres”. From my point of view, the topic is fascinating. The manuscript is concise and well-written. However, the manuscript has some issues which need to be addressed before its publication in Marine Drugs. Following are my concerns:

1)      The literature on using chitosan nanoparticles for BDNF delivery has not been discussed in detail. Authors should discuss more articles on BDNF delivery by chitosan nanoparticles.

2)      Authors should also discuss the novelty aspect of their work.

3)      The authors should also write the product number, company name, and exact quantity of all the chemicals they have used.  

4)      For the comparison, the magnification of the confocal images should be the same for different time points.

5)      Why neuritic growth velocity decreased in the case of chitosan and chitosan+BDNF but not in the case of BDNF only?

6)      What about the cytotoxicity of the chitosan microsphere used in the study?

Author Response

Thank you for reviewing our work and your comments on the manuscript. I will answer your questions in the same order that were exposed:

1, 2) Following the reviewer's instructions, we included a paragraph in the Discussion section discussing the delivery of BDNF with respective references; and the novelty of the work is declared in Discussion and Conclusions sections.

3) The product number, company name, and quantity of all the chemicals has been described in Materials and Methods and revised in all text.

4) The microscopic magnification is different at 72h and 120h because untreated PC12 cells proliferated and clumped together. A wider microscopic field and lower magnification were required at 120 h than at 72 h to include a similar number of cells to compare neuritic growth.

5) The following text was added to the text in Discussion section that answer the question of reviewer

There was a reduction in neuritic growth rate in the case of chitosan-BDNF microspheres, but not for free BDNF treatment (Figure 2h). The kinetic release of proteins from the chitosan particles reaches a maximum around the third day and is maintained on successive days. Consequently, a high concentration of local BDNF released from chitosan particles promotes the rate of neuritic growth; but PC-12 cells are close to the growth limit and then the growth rate slows down. In the case of free BDNF, the rate of neuritic growth is almost the same because the BDNF concentration did not change in solution.

6) Regarding cytotoxicity of chitosan microsphere. We include an MTT viability assay as supplementary material. Here we shown that presence of chitosan microsphere did not affected the metabolism of PC12 cells.

Thanks for your attention,

The authors

Reviewer 2 Report

Manuscript No.: marinedrugs-2153515-peer-review-v1

Title: Fast and Sustained Axonal Growth by BDNF Released from Chitosan Microspheres

Marine Drugs

Reviewer's Decision: Accept after major revision

The authors of this research work describe the fast and sustained release of Brain-derived neurotrophic factor from the chitosan microsphere. The research is significant and should be published in the Marine Drugs. However, the manuscript must be significantly improved before it can be published. As a result, I recommend accepting the manuscript after significant and satisfactory revisions. The following are the detailed comments:

1.     Title: The title appears confusing, as no abbreviation should be mentioned is not suitable in the title. The title of any research study is to the point so that the reader may understand the maximum idea of the research. Therefore, removing the title is recommended to avoid confusion for the readers.

2.     Abstract: The abstract is a comprehensive summary of the whole research article. The abstract contains numerous grammatical and formatting errors. It is suggested to improve the grammar and English language problems. The abstract section is more introductive and contains methodology information. They should reduce the informative and methodology section and add more results outputs with specific biomedical applications. The incomplete information in the abstract may confuse the readers.

3.     Introduction: The introduction section should address the historical and current trends supported by the literature. The introduction and the rest of the manuscript have several grammatical and formatting errors. Please improve the grammar, language, and formatting issues in the manuscript.

4.     References: The manuscript lacks the literature citation of some highly interesting, most recent relevant works; thus, the references are not up to date. These citations will help to explain chitosan behavior in the introduction, results, and discussions. In this regard, the author should refer to some of the most recent papers on hydrogel, such as

·       Yaqoob, Z., et. al. (2021). Chitosan/poly vinyl alcohol/graphene oxide based pH-responsive composite hydrogel films: drug release, anti-microbial and cell viability studies. Polymers13(18), 3124.

·       Haider, A, et. al. (2021). Smart and pH-sensitive rGO/Arabinoxylan/chitosan composite for wound dressing: In-vitro drug delivery, antibacterial activity, and biological activities. International Journal of Biological Macromolecules192, 820-831.

5.     Materials and methods: This section is missing, and please add it; otherwise, it may confuse the readers.

·       The author has used the same "degree" sign for temperature and angle. It is recommended to use the "degree" symbol accordingly throughout the manuscript.

6.     Results and Discussions: The following issue must be taken into consideration.

a.     Use the proper minus sign in table 1 rather than a dash.

b.     The figure caption and style should be identical throughout the manuscript to keep the continuity of the work.

7.     Conclusions: The conclusion section is the most important summary of a research article, and it should be based on the conclusion for the conclusion. It is recommended to add a conclusion section by comparing the best result output.

8.     As per the comments given for the results and description.

In summary, the reported work has significant value; however, a major and thorough improvement/correction of language, grammar, syntax, etc., is necessary to improve the paper's quality and make it publishable in the Marine Drugs.

·       All the abbreviations should be defined before their 1st-time use.

Author Response

Thank you for reviewing our work and your comments on the manuscript. I will answer your questions in the same order that were exposed:

1) Following the reviewer's observation, I checked PubMed for the use of BDNF in the title of the articles. I have noticed that it is used with a similar frequency to the full name, some recent examples are:

_ Genetic Val66Met BDNF Variant Increases Hyperphagia on Fat-rich Diets in Mice. Endocrinology. 2023 Jan 12:bqad008. doi: 10.1210/endocr/bqad008. Online ahead of print.

_ Fasting for 20 h does not affect exercise-induced increases in circulating BDNF in humans. J Physiol. 2023 Jan 11. doi: 10.1113/JP283582. Online ahead of print.

Also in this journal BDNF appears in the title of some articles:

_ Neoagaro-Oligosaccharides Ameliorate Chronic Restraint Stress-Induced Depression by Increasing 5-HT and BDNF in the Brain and Remodeling the Gut Microbiota of Mice

Mar. Drugs 2022, 20(11), 725; https://doi.org/10.3390/md20110725 - 18 Nov 2022

_ ω-3 DPA Protected Neurons from Neuroinflammation by Balancing Microglia M1/M2 Polarizations through Inhibiting NF-κB/MAPK p38 Signaling and Activating Neuron-BDNF-PI3K/AKT Pathways

Mar. Drugs 2021, 19(11), 587; https://doi.org/10.3390/md19110587 - 20 Oct 2021

2) Following instructions of reviewer the Abstract section was revised and corrected, some experimental data (buffer solutions, time of incubations) were removed and biomedical application were added.

3, 4) The Introduction was revised and corrected and suggested references were added

5) The Materials and Methods section appears after the Discussion section.

6 a) The minus symbol was inserted

6 b) Regarding figure caption, microscopic magnification is different at 72h and 120h because untreated PC12 cells proliferated and clumped together. A wider microscopic field and lower magnification were required at 120 h than at 72 h to include a similar number of cells to compare neuritic growth.

7) The Conclusions section is now included

Thanks for your attention,

The authors

Reviewer 3 Report

Introduction

1. The authors should add the art of the previous literature including the mechanism of BDNF clearance, protection, and prolongation of its circulation.

2. The introduction section lacks the role of chitosan in minimizing protein opsonization.

Results

1. Where are the results of the Encapsulation of BDNF in chitosan microspheres?

2. In lines 88-89, add a reference.

3. Section 2.3. BDNF-chitosan microspheres promote significant neural cell differentiation and neuritic 91 out-growth in PC12 cells.

Change the section title to Change title to:

Effect of BDNF-chitosan microspheres on neural cell differentiation and neuritic growth in PC12 cells. 

Materials and Methods

1. Define the molecular weight of chitosan, and add the full names of DMEM, sodium acetate, and acetic acid.

2. For the encapsulation procedures, what was the atomization equipment used, and its specifications?

3. Lines 174-175, add a reference for the entrapment study.

4. Lines 177-182, Restructure this section with referring to the addition of chitosan-loaded BDNF microspheres.

General Comment

Add a conclusion for the study.

Author Response

Thank you for reviewing our work and your comments on the manuscript. I will answer your questions in the same order that were exposed:

Introduction

1, 2) Following instructions of reviewer the literature regarding BDNF clearance, protection and its circulation; and its role avoiding protein opsonization were added.

Results

1) This study describes the BDNF encapsulation procedure in the Materials and Methods section. We tried to follow the release of BDNF from the chitosan microspheres by specific ELISA; but the optical density values were in the limit detected by this ELISA. So, we decided to use the biological activity of BDNF on PC12 to select the best encapsulation procedure. However, and in agreement with reviewer, we are aware that direct neurotrophin detection and release kinetic experiments must be performed.

2, 3) A reference was added and the title was changed.

Conclusions. This section was included

Materials and Methods

1, 2)   The molecular weight of chitosan, the full names of DMEM, sodium acetate and acetic acid were included. Spray equipment and specifications are included.

3,4) We describe chitosan-BDNF microsphere treatment of PC12 cells and immunocytochemistry in separate sections. No treatment with fixed cells was performed.

Thanks for your attention,

The authors

Round 2

Reviewer 1 Report

Authors have made significant changes in the manuscript. The manuscript can be accepted in its current form.

Author Response

Thanks to the reviewer for his comments on the manuscript and favorable decision on this work.

Regards, 

The authors

Reviewer 2 Report

All the comments have been addressed and the manuscript can be accepted in the present form.

Author Response

(The authors gave the same response as above.)

Reviewer 3 Report

In the results section 2.1., Where are the results of the Encapsulation of BDNF in chitosan microspheres?

Section 4.5. Chitosan microspheres-BDNF treatment, add a reference for the procedure!

Line 207, Hoechst agent (Sigma-Aldrich Chemie, Steinheim, Germany), should added to materials section

Author Response

Thanks to the reviewer for this second round of review of our work and his comments on the manuscript. I will answer your questions in the same order in which they were asked:

1) In relation to the results of the encapsulation of BDNF in chitosan microspheres. We have not reached conclusive data in this regard, so we do not present them here. But, as I answered before, we are aware of this and some experiments that increase the amount of protein during encapsulation were designed to detect neurotrophin by specific ELISA or Western blot procedure.

2) A sentence has been added to section 4.5, with the reference [30], for the treatment of cells in cell culture.

3) Hoechst agent (Sigma-Aldrich Chemie, Steinheim, Germany), was included in materials and methods, section 4.6.

Thanks again for your attention,

The authors